# A Reinforcement Learning Approach to Speech Coding

Jerry Gibson *,† and Hoontaek Oh †

Department of Electrical and Computer Engineering, University of California,
Santa Barbara, CA 93106, USA; hoontaek@ucsb.edu
* Correspondence: gibson@ece.ucsb.edu
† These authors contributed equally to this work.

**Abstract:** Speech coding is an essential technology for digital cellular communications, voice over IP, and video conferencing systems. For more than 25 years, the main approach to speech coding for these applications has been block-based analysis-by-synthesis linear predictive coding. An alternative approach that has been less successful is sample-by-sample tree coding of speech. We reformulate this latter approach as a multistage reinforcement learning problem with $L$ step lookahead that incorporates exploration and exploitation to adapt model parameters and to control the speech analysis/synthesis process on a sample-by-sample basis. The minimization of the spectrally shaped reconstruction error to finite depth manages complexity and serves as an effective stand in for the overall subjective evaluation of reconstructed speech quality and intelligibility. Different control policies that attempt to persistently excite the system states and that encourage exploration are studied and evaluated. The resulting methods produce reconstructed speech quality competitive with the most popular speech codec utilized today. This new reinforcement learning formulation provides new insights and opens up new directions for system design and performance improvement.

**Keywords:** reinforcement learning; speech coding; exploration; exploitation; dual control

## 1. Introduction

Speech coding was a key technology enabling the evolution of digital cellular communications to what it is today. Current speech codecs are block-based analysis-by-synthesis linear predictive coders and their design draws on a fortuitous match between voiced speech and an autoregressive model, digital signal processing, and a deep knowledge of the speech signal and human perception.

In this paper, we recast the speech coding problem as a reinforcement learning problem [1]. We build this new reinforcement learning speech coding structure around the well-known recursive tree coding of speech methods that have been heretofore motivated by rate distortion theory [2,3] and designed by mostly cut-and-try techniques based on domain knowledge [4–6]. In recent years, reinforcement learning approaches have been investigated as part of machine learning or deep learning approaches; however, reinforcement learning can be accomplished without being associated with neural networks. We do not utilize neural networks here.

Recursive tree coding requires choosing an excitation for a model with unknown parameters that attempts to track the input speech on a sample-by-sample basis subject to a bit rate constraint and to reproduce the speech at a remote location (the Decoder) subject to a perceptual distortion measure. Stated in this way, this codec structure maps rather directly onto reinforcement learning, since not only does the excitation have to be selected to control the reconstruction at the Decoder, but it has to drive the adaptive identification of the unknown model parameters [7]. Thus, the ideas of exploration (adaptive model parameter identification) and exploitation (produce a suitable reconstruction at the Decoder) from reinforcement learning naturally capture the dual role played by the selection of an appropriate excitation [1,8–10]. Of course, this dual control requirement was foreseen by

Feldbaum in the 1960s as a problem in stochastic control, wherein the control had the dual role of probing (exploration), to learn model parameters, and caution (exploitation) to control the system carefully knowing that there are unknown parameters [7,11–13].

By developing and analyzing this reinforcement learning method, we demonstrate that these new speech coding structures offer competitive performance with the standardized, widely used Adaptive Multirate Narrowband (AMR-NB) speech codec, motivate new speech codec designs, and offer new insights and research directions for the speech coding problem.

The paper is organized as follows. Section 2 provides a brief overview of the most recent and most deployed speech codecs, specifically pointing out their structure so they can be contrasted with the current approach. The terminology used in the paper as it relates to the reinforcement learning and stochastic control formulations is presented in Section 3. Section 4 describes how the speech coding problem can be viewed and set up as a reinforcement learning problem. The several adaptation algorithms for identifying and updating the parameters of the system model are presented in Section 5. Section 6 presents and discusses the needed spectral error shaping so that the learning and control functions optimize a meaningful cost function. The particular Value Function is discussed and justified in Section 7. The several control policies investigated are developed and motivated in Section 8, and the form of the lookahead cost function is explained in Section 9. Section 10 contains the experimental results of putting all of the components together and shows how exploration and exploitation is used to improve speech codec designs, and compares the reinforcement learning speech coding performance with the AMR-NB codec described in Section 2. Conclusions are contained in Section 11, which succinctly summarizes the method, the insights provided, and future research directions.

The appendices provide background information and elaborate details that do not fall in the main narrative development of the reinforcement learning approach. In particular, Appendix A provides concise descriptions of differential pulse code modulation and tree coding, which serve as a context for our current reinforcement learning method. Appendices B and C contain additional well-documented details about the adaptation of the longer term predictor, and Appendix D summarizes the voice activity detection and comfort noise generation techniques employed to separate silence and Unvoiced speech from Voiced speech segments.

## 2. Current Standardized Speech Codecs

The most widely installed speech codec today is the Wideband Adaptive Multirate (AMR-WB) speech codec [14]. It has been standardized for almost 20 years (since 2001, a narrowband version since 1999) and offers several transmitted bit rates from 6.6 kilobits/s (kbits/s) to 23.85 kbits/s that can be chosen depending on the channel quality. The one way algorithmic delay is 26 ms and it has been tested across many languages. The studies on neural networks for end-to-end speech coding thus far have used AMR-WB as a comparison codec.

A newer speech codec is the Enhanced Voice Services (EVS) codec that was standardized by 3GPP in September 2014. This codec offers enhanced quality for both narrowband and wideband speech, enhanced quality for mixed speech and music content, robustness to packet loss and delay jitter, and backward compatibility with AMR-WB [15]. The EVS codec was tested for 10 languages. At 9.6 kbits/s, EVS significantly outperforms AMR-WB at 23.85 kbits/s.

Both of these successful codecs, and their predecessors in the past 35 years [16,17], are based on taking in a frame or block of speech and using block-based least squares to calculate the coefficients of a linear prediction model. The best fixed excitation for the linear prediction model is then found from a sparse codebook in combination with an adaptive method to model the pitch excitation via analysis-by-synthesis (AbS) over the block of speech to minimize a perceptually weighted squared error. These parameters are then coded for transmission to the receiver to reconstruct the block of speech using a

linear prediction model followed by post shaping the decoder output. The process is then repeated for the next block of input speech [18].

### 3. Reinforcement Learning and Stochastic Control: Terminology and Methods

Reinforcement learning has a longer history than one might imagine and has close ties with stochastic control and dynamic programming [1,8–10], and less directly with system identification [19]. Since our system and methods do not involve neural networks, we use a mix of the reinforcement learning and stochastic control nomenclature and notation. We have the four main elements of reinforcement learning [1]:

- A model of the environment that we call the System Model;
- A control policy that describes the behavior of the learning agent;
- A cost function instead of a reward function;
- A cost-to-go that tries to emulate a Value Function.

The cost-to-go attempts to capture the true performance of the system as measured against the Environment, but it is only an approximation, and the actual performance is evaluated after the entire trajectory is completed. This performance evaluation involves estimating the quality and perceptability of any errors over the full lifetime of the interaction with the Environment after the fact. This performance evaluation constitutes our Value Function as discussed in Sections 6 and 7. This after the fact Value Function appears to be unique to the reinforcement learning literature.

Notationally, we generally adopt that of stochastic control because we have a cost function rather than a reward [8], and we are not operating within a neural network construct. We sense the Environment by an observation of the system state, $s(k)$; we have a control policy that generates a sequence $u(k)$; we measure the error at stage $k$ by a cost function $J_k$; we approximate the cost-to-go by $\sum J_k$. The System Model maps the control sequence and current state of the System Model to an estimate of the state of the Environment.

Conceptually, our work is motivated by the dual control theory formalism of Feldbaum [11–13] where the control sequence must be used not only to control the system but also to perturb the system in order to learn system parameters; however, we do not have nor utilize the probabilistic knowledge assumed in dual control. Wittenmark [7] has a good overview with additional references.

In dual control, the actions of the control are characterized as probing or caution, depending upon whether the control is being used to perturb the system to learn parameters or the control is driving the system cautiously since it realizes there are unknown parameters [7]. Here we adopt the reinforcement learning terminology of exploration (probing) and exploitation (caution) to describe the control actions.

### 4. Speech Coding as Reinforcement Learning

Before we develop the reinforcement learning structure for speech coding, we need to outline the basic speech coding problem. The idea is that the speech signal (the Environment) can be observed at the Encoder and the operations in the Encoder must come up with a low bit rate representation that can be sent over a digital communications channel to the Decoder in order to reconstruct the speech. Note that the bit rate constraint actually can map directly into an energy or power constraint, which are the kinds of constraints usually placed on control sequences in stochastic control. Although the bit rate constraint can seriously limit the representation that can be sent, the reconstructed speech quality and intelligibility must be sufficient to satisfy the User; therefore, there is an Encoding process that generates a bit rate sequence that is used at the Decoder to reconstruct the speech. We start the discussion with a development of the Encoding process and then specify the Decoder that reconstructs the speech.

A block diagram of the speech encoding process as a reinforcement learning problem is shown in Figure 1. The Environment or System is human speech so $s(k)$ represents the input speech samples, and the goal of the Reinforcement Learning System is to develop

a control sequence $u(k)$ that will drive a system model to approximate the input speech subject to a rate constraint on the number of bits per second that is used to represent $u(k)$. The rate constraint is an implicit indication that the System Model to be controlled is actually the Decoder in the User's mobile phone at a remote location.

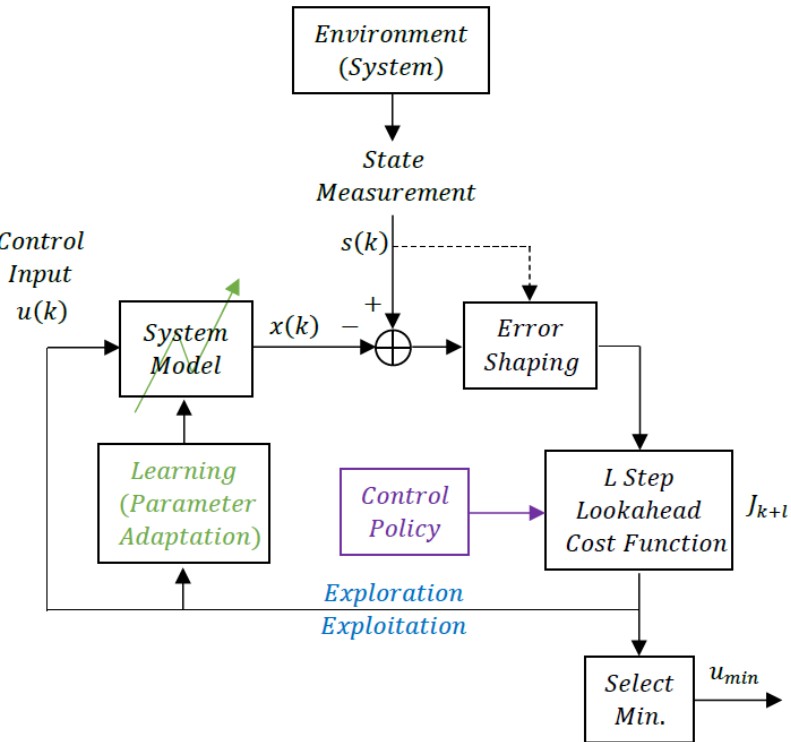

**Figure 1.** Reinforcement learning for speech coding.

The problem setup thus requires the development of:

- A System Model;
- A Cost Function;
- A Cost-to-Go function;
- A Control Policy.

We begin with the System Model.

### 4.1. The System Model

From decades of accumulated domain knowledge of the speech modeling process, we know that a reliable model for speech sequences is a discrete autoregressive (AR) process

$$s(k) = \sum_{i=1}^{M} a_i s(k - i) + w(k) \tag{1}$$

where the autoregressive parameters $a_i, i = 1, 2, ...M$, are called the linear prediction coefficients in speech processing applications, and $w(k)$ is the excitation sequence [18]. The excitation can be a periodic sequence or a noise sequence or a mix of the two. The excitation sequence can thus be modeled as

$$w(k) = \sum_{i=-1}^{1} p_i w(k - d + i) + \epsilon(k) \tag{2}$$

where $d$ is the long term pitch redundancy and $\epsilon(k)$ is some random driving sequence. The AR coefficients $a_i, i = 1, 2, \ldots M$ are time-varying as are the excitation signal characteristics

$d$, $p_i$, $i = -1, 0, 1$, and $\epsilon(k)$. All of this knowledge of the Environment can be used to construct the System Model to be controlled and the various adaptive identification algorithms.

The System Model to be controlled cannot include actual speech samples since they are not available at the remote Decoder, and are, in fact, what we are trying to reproduce using our control sequence. Furthermore, even if the System Model were exact, we do not know the various parameters, which must be adaptively identified, but based on Equations (1) and (2), we can write that the System Model at least should include the structure

$$x(k) = \sum_{i=1}^{M} \widehat{a}_i x(k-i) + \sum_{i=-1}^{1} \widehat{\beta}_i u(d+i) + u(k) \tag{3}$$

where the hats $\widehat{a}_i$ and $\widehat{\beta}_i$ indicate estimated parameters and $u(k)$ is the control sequence.

This model needs to be modified further in order to capture the fact that the AR and long term redundancy parameters are being adaptively estimated based on noisy measurements, that is, not the input speech itself, since the input speech is what we are attempting to reproduce at the Decoder.

Research has shown that when estimating AR coefficients based on noisy measurements, the recursion should include a moving average (MA) component in addition to the AR component [20]. In addition, it is a less well-known result that if the AR coefficients are being estimated within the prediction recursion loop, the coefficient adaptation can start to track itself and not the observed input speech, but the inclusion of a MA component can help prevent this from happening [21].

Incorporating the MA part, Equation (3) becomes

$$x(k) = \sum_{i=1}^{N_a} \widehat{a}_i x(k-i) + \sum_{i=-1}^{1} \widehat{\beta}_i u(d+i) + \sum_{i=1}^{N_b} \widehat{b}_i u(k-i) + u(k) \tag{4}$$

Equation (4) constitutes the System Model to be controlled. This System Model also requires the estimation/identification of the $\widehat{a}_i$, the $\widehat{\beta}_i$, the pitch lag $d$, and the $\widehat{b}_i$. Since the adaptive algorithms to identify these parameters are also driven by the control sequence $u(k)$, we have a dual control or reinforcement learning problem. The adaptive identification algorithms for estimating these parameters are described in Section 5.

The critical importance of Equation (4) and the recognition that the control $u(k)$ not only drives the reconstruction but also simultaneously drives the parameter estimation/identification algorithms is the essence of this contribution. Both adaptive DPCM (ADPCM) and tree coders can have adaptive parameter estimation algorithms but the excitation in these systems is wholly chosen to achieve good reconstruction. More specifically, in ADPCM the prediction error is simply quantized to yield a quantized prediction error that most closely approximates the quantizer input. The ADPCM system does not choose the quantized value with the knowledge that this value will be driving the several parameter estimation/system identification algorithms. Similarly for tree coders, the excitation is chosen to approximate the best reconstruction to some depth $L$ lookahead, but the choice of excitation does not reflect the fact that it is also driving the parameter estimation and system identification algorithms.

The reinforcement learning/stochastic control formulation directly captures that dual role of the control sequence (the excitation sequence in ADPCM and tree coding) and thus leads the designer/researcher to realize that the control policies should take advantage of both roles. The remainder of the paper, particularly, Sections 8 and 10 elucidate both functions of $u(k)$ and the performance gains possible.

## 5. Learning (Parameter Adaptation)

From Equation (4), we see that the parameters to be estimated or identified are $\widehat{a}_i$, $\widehat{\beta}_i$, lag $d$, and $\widehat{b}_i$, and note that only signals available at the Decoder as well as the Encoder can be used in their calculation. We employ different algorithms and approaches for

each parameter. For the parameters of the AR component of Equation (4), $\widehat{a}_i$, we utilize a Recursive Least Squares (RLS) lattice algorithm because of its fast convergence and stability properties [22,23]. For the parameters of the MA component, $\widehat{b}_i$, we utilize a very simple polarity driven gradient algorithm taken from ITU-T G.726/727 [24,25]. The calculation of the lag $d$, uses a combination of a block-based method using only past model values jointly with a recursive, gradient-like update [26]. The coefficients for the long term prediction component, $\widehat{\beta}_i$, are propagated based on gradient algorithms [26]. The details for all of these are given in the following subsections.

*5.1. AR and MA Parameter Adaptation*

The autoregressive predictor coefficient adaptation is based on a lattice structure as shown in Figure 2.

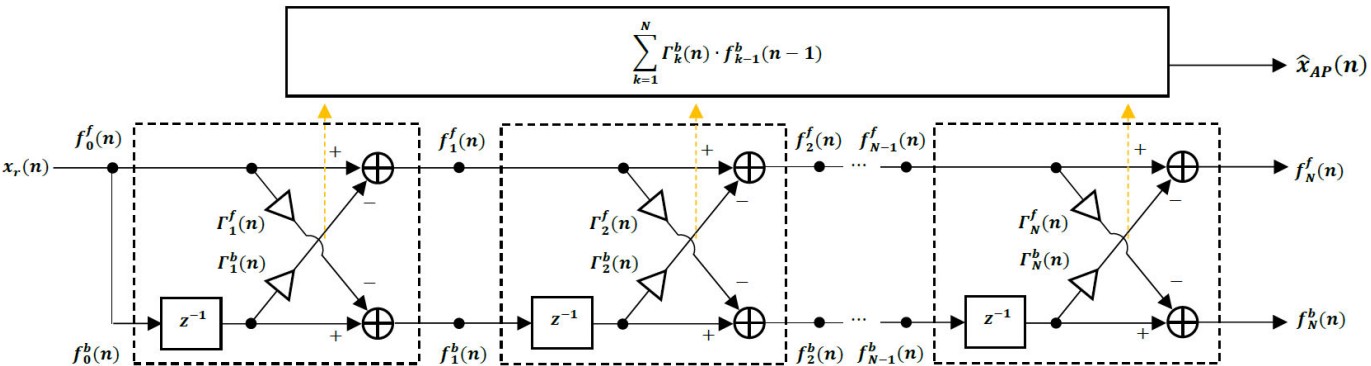

**Figure 2.** Lattice Predictor.

The adaptation of the parameters in this lattice predictor is accomplished using the Recursive Least Squares (RLS) lattice algorithm, which operates by first defining the initial conditions for $n$ = 0 as [22,23,27]

$$f_0^f(n) = f_0^b(n) = x_r(n) \tag{5}$$

$$\zeta(n) = 0 \tag{6}$$

$$E_1^f(n) = E_1^b(n) = \lambda E_1^f(n-1) + [f_0^f(n)]^2 \tag{7}$$

Then, we perform the following recursions in order for $k$ = 1, 2, . . . , N,

$$\Gamma_k(n) = \lambda \Gamma_k(n-1) + \frac{f_{k-1}^f(n)f_{k-1}^b(n-1)}{1 - \zeta_{k-1}(n)} \tag{8}$$

$$\Gamma_k^f(n) = \frac{\Gamma_k(n)}{E_k^f(n)} \tag{9}$$

$$\Gamma_k^b(n) = \frac{\Gamma_k(n)}{E_k^b(n-1)} \tag{10}$$

$$E_{k+1}^f(n) = \frac{E_k^f(n) - \Gamma_k^b(n)\Gamma_k(n)}{v^2} \tag{11}$$

$$E_{k+1}^b(n) = \frac{E_k^b(n-1) - \Gamma_k^f(n)\Gamma_k(n)}{v^2} \tag{12}$$

$$\zeta_k(n) = \zeta_{k-1}(n) + \frac{[f_{k-1}^b(n-1)]^2}{E_k^b(n-1)} \tag{13}$$

where the forward and backward prediction errors are updated as

$$f_k^f(n) = f_{k-1}^f(n) - \Gamma_k^b(n) f_{k-1}^b(n-1) \tag{14}$$

$$f_k^b(n) = f_{k-1}^b(n-1) - \Gamma_k^f(n) f_{k-1}^f(n) \tag{15}$$

The $0 < \lambda < 1$ and $0 < v < 1$ parameters are selected to keep the algorithm from reaching an undesirable steady state as the number of samples processed becomes larger, and thus make sure that the algorithm continues to adapt and track the rather rapidly changing input speech signal.

Advantages of the RLS lattice algorithm are that the recursions are identical for each stage, all of the variables are scalars, and it exhibits fast convergence, so it allows the coder to respond quickly to changes in the speech signal [22,23]. Honig and Messerschmitt [22] provide extensive derivations and discussions of least squares lattice algorithms.

The $N_b$ MA coefficients $(\widehat{b}_i)$ of the predictor are adapted as [24,25]

$$\widehat{b}_i(k) = (1 - 2^{-8})\widehat{b}_i(k-1) + 2^{-7}\text{sgn}[u(k)]\text{sgn}[u(k-i)] \tag{16}$$

for $i = 1, 2, \ldots, N_b$. This algorithm is obviously extremely simple but plays an important role by aiding in the overall adaptation performance, as discussed in Section 4.1, while avoiding any stability issues.

*5.2. Quasi-Periodic Excitation Adaptation*

The quasi-periodic excitation consists of a three coefficient component centered around the estimated pitch lag $d$ and is of the form [26,28,29]

$$\widehat{x}_{PE}(n) = \sum_{k=-1}^{k=+1} \beta_k x_r(n - d - k) \tag{17}$$

The calculation of the quasi-periodic pitch lag $d$ and the weighting coefficients $\beta_k$ uses a combined block and recursive method since the recursive method alone based on the output signal can have difficulty tracking a rapidly changing speech signal over a longer time period.

The way to calculate the pitch lag for each sample is to find the maximum of the autocorrelation function $\hat{\rho}(m)$ [26,28,29]. After initializing the pitch lag using a block of past *outputs* , as indicated in Appendices B and C, it is adapted on a sample-by-sample basis. The estimate of the autocorrelation function at lags of $m = d_k + 1, d_k$, and $d_k - 1$ for each sample is used for tracking. The estimated autocorrelations $\hat{\rho}(m)$ are obtained from the following [26,29] recursion using the output of the pitch excitation $e(k)$:

$$\hat{\sigma}_e^2(k) = \eta\hat{\sigma}_e^2(k-1) + (1-\eta)(e(k))^2 \tag{18}$$

$$\hat{\rho}^{(k)}(m) = \eta\hat{\rho}^{(k-1)}(m) + (1-\eta)\frac{e(k)e(k-m)}{\hat{\sigma}_e^2(k)}, \tag{19}$$

where $\hat{\sigma}_e^2$ is the estimated variance of $e$ and $\eta = 0.95$. After updating the estimated autocorrelation function, the pitch lag $d_k$ is updated to be $d_k + 1$ if $\hat{\rho}^{(k)}(d_k + 1)$ is a maximum, $d_k - 1$ if $\hat{\rho}^{(k)}(d_k - 1)$ is a maximum, or $d_k$ otherwise.

If the pitch lag is adjusted, then the values of the estimate of the autocorrelation function are shifted in the same way that the weighting coefficients are shifted. The new autocorrelation value, either $\hat{\rho}^{(k)}(d_k - 1)$ or $\hat{\rho}^{(k)}(d_k + 1)$, is computed to be a constant fraction of $\hat{\rho}^{(k)}(d_k)$, typically 0.3.

## 6. Error Shaping

For speech coding, the time domain mean squared error is a poor indicator of how well the overall system is performing with or without lookahead; however, perceptually

shaping the error in the frequency domain before the sum of the squared error calculation can yield good performance and is computationally tractable. The error shaping employed in this work has the form

$$W(z) = \frac{1 - \sum_{k=1}^{8} a_k \mu^k z^{-k}}{1 - \sum_{k=1}^{8} a_k \gamma^k z^{-k}} \tag{20}$$

where the $a_k, k = 1, 2, \ldots, 8$ are the AR predictor coefficients calculated based upon a block of the input speech, that is, the observed state of the Environment, and $\mu$ and $\gamma$ are constants to be selected. Although various combinations of $\mu$ and $\gamma$ are possible, their values should fall in the interval between 0 and 1, and based on experiments, we set the parameters $\mu = 1$ and $\gamma = 0.8$. Other values of $\mu$ and $\gamma$ can be and are used in block-based CELP codecs, and the current codec is not particularly sensitive to variations in the values chosen here. More specifically, the desired spectral error shaping is not achieved by fixed values of $\mu$ and $\gamma$ across all speech segments, but adaptive methods for these parameters have not been determined in prior experiments for tree coders or for the standardized CELP based codecs.

The calculation of the perceptual error filter coefficients in Equation (20) uses a block-based method on the input speech samples since error shaping is only performed at the Encoder, and there is no need to transmit the perceptual weighting filter coefficients to the Decoder. We calculate the model coefficients from the input speech frames by using nonoverlapping rectangular windows of the 200 future input samples with an update rate of every 200 samples. This corresponds to a latency of 25 ms at 8000 samples/s and constitutes the maximum encoding delay associated with the current codec. We have experimented with using 100 past samples and 100 future samples in this process and the results are very similar. To reduce latency, this parameter estimation process for error shaping deserves further attention.

In the time domain, we denote the spectrally shaped weighted error sequence by $e_w(k) = [s(k) - x(k)]_w$, where $x(k)$ and $s(k)$ are as in Figure 1, so that the cost-to-go function is $\sum J_k = \sum e_w^2(k) = \sum J_k = \sum [s(k) - x(k)]_w^2$ where the sum is over $L$ time instants starting with the current instant $k$.

The perceptual error shaping filter provides only an imperfect and approximate indicator of the actual quality and intelligibility of the reconstructed speech, but it is critically important to obtaining acceptable system performance. The goal of the perceptual weighting filter is to keep the reconstruction error spectral envelope below that of the input speech envelope across the frequency band of interest, since this allows the speech spectrum to mask the error spectrum. An example is given in Figure 3.

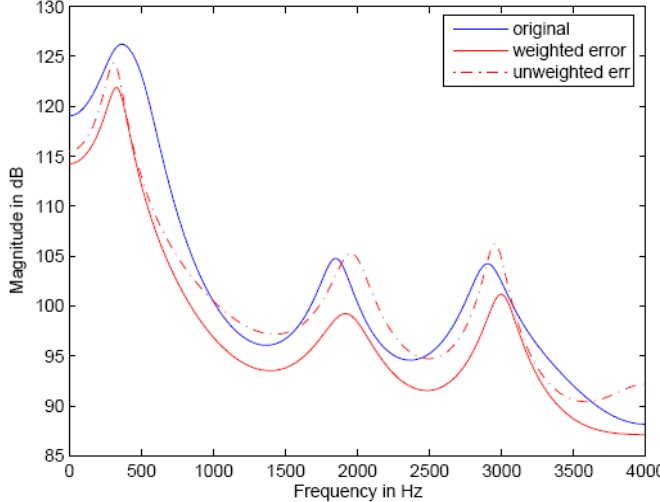

**Figure 3.** Perceptual spectral error shaping example.

In Figure 3 we see that the unweighted error spectral envelope falls above the input speech spectral envelope at some frequencies and the perceptual error weighting is able to push the error envelope down below that of the input speech; however, the error spectral envelope becomes very close to the input spectral envelope at 2000 Hz and just above 3000 Hz. The most desirable result would be for the error spectral envelope to be a constant amount below the input spectral envelope across the entire band. This is difficult to achieve, and it is well known that even with the error spectral shaping, the envelope of the error spectrum may not fall below that of the input spectrum.

However, this is the best known approach available and the weighted squared error is important, even though imperfect, in reproducing a pleasing reconstruction at the Decoder. The approximate Cost-to-Go, $\sum J_k$, is the sum to some depth $L$ of perceptually shaped squared errors in the reconstruction.

## 7. The Value Function

The goal of the overall control problem is to reproduce speech at the Decoder that sounds as close to the original input samples as possible. This performance evaluation can only be determined by human listening tests or objective performance criteria that are representative of human listening tests as captured by ITU-T standard software that compares the input speech to the reconstructed sequence [30], after the entire utterance has been reconstructed. We use the ITU-T standard software [30] as the final performance indicator (Value Function) in this work, which generates the PESQ-MOS, a quantity between 0 and 5, higher values being preferred. We call the PESQ-MOS the Value Function. This objective comparison of the full input utterance with the full reconstructed utterance is a strong indicator of perceptual quality and intelligibility similar to what would be obtained with large-scale human listening tests. The latter are too expensive and too difficult to organize and too complicated to conduct except in the validation of a final design for establishing an International Standard. We note that all of the speech samples used in this paper are taken from the ITU-T P.501 standard [31].

A PESQ-MOS value of 3.5 or greater is desired, even when the entire utterance including the VAD/CNG, which is used for silence intervals, is compared to the original utterance. See Appendix D for a discussion of VAD/CNG. A PESQ-MOS value of 3.5 or better is associated with good quality, intelligible speech and is about the performance obtained in general use by cell phones; however, since our Reinforcement Learning method is only used for Voiced speech, the most important Value Function is the PESQ-MOS over the Voiced segments only. Both Value Functions are used for performance evaluation in the following. Note also that the PESQ-MOS scores are coupled with the bit rate, equivalent to the control sequence energy, in order to obtain a meaningful Value Function. This coupling is necessary because an higher bit rate constraint, which corresponds to an higher constraint on control sequence average energy, should always lead to better performance, reflected here as PESQ-MOS.

## 8. Control Policy

A large part of this investigation, and one that highlights the power of the reinforcement learning approach to speech coding, is the selection of a Control Policy. We impose an average rate constraint of 1.5 bits/sample during active voice regions and code the Silence regions with a much simplified, relatively standard approach. The active voice regions are thus of primary interest in this paper. The bit rate constraint corresponds to an average energy or average power constraint somewhat directly, since an increased speech coding bit rate requires a higher transmitted bit rate over the channel. Because of this direct mapping, rather than bring in the modulation and communications ideas, we focus on speech compression since many different techniques can be used for transmission over a physical channel.

*8.1. Control Tree Sequences*

There are a host of possible control policies but we consider tree structured control sequences since they provide a direct connection to bit rate and also allow adaptive gain calculations for adjusting to the varying amplitude of the speech samples in the Environment from the transmitted/optimized control sequence without using extra energy or bits to send the gains to the receiver. To produce an average bit rate of 1.5 bits per sample, we interleave 4 level, or 2 bit, and 2 level, or 1 bit, trees in equal amounts.

An example 4-2 control tree segment is shown in Figure 4.

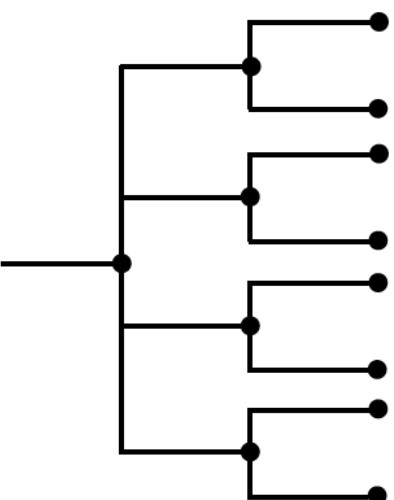

**Figure 4.** A 4-2 multitree control segment.

If this multitree control sequence is just alternated as 4-2 repeatedly; the rates of the interleaved control trees would be $R = 2$ bits/symbol and $R = 1$ bit/symbol, so the average rate would be $R_{avg} = 1.5$ bits/symbol, thus meeting the rate or energy constraint; however, since imposing this periodic structure on the control sequence is rather arbitrary, other approaches are investigated.

Utilizing the same two four-level and two-level trees, we randomize between choosing each with each tree being equally likely, thus again meeting the bit rate or energy constraint. The specific approach taken here is to impose a fixed rate of 1.5 bit per sample on a length 40 sequence of interleaved four level and two level trees, where the four and two level trees are equally likely, and generate all possible such 40 sample sequences. These length 40 sequences are then concatenated over the length of the utterances and the best control sequence from that group with the highest average PESQ-MOS over all sentences is selected. These experiments were conducted using the utterances F1, F2, M1, and M2. This is called the 4-2 randomized control sequence and is shown in Figure 5.

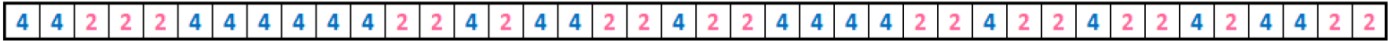

**Figure 5.** The 40 sample 4-2 randomized control multitree sequence.

To ensure that there is exploration in addition to exploitation, another control sequence is investigated. This control sequence injects a 5-level tree segment at the beginning of each Voiced speech portion of an utterance, with the two outer levels being a nonuniform, higher level excitation value. The 5-level control tree segment is symmetric about the origin so only the zero level and positive levels are shown in Figure 6.

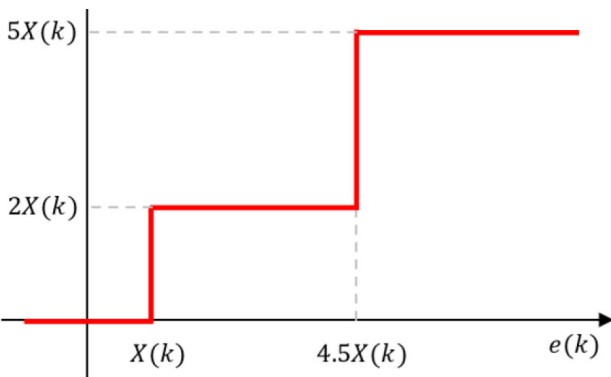

**Figure 6.** The 5-level control tree segment.

We see that, unlike the four-level control segment, which has a uniform distance between output points, the 5-level control segment has an outer level that is two and a half times larger than the inner level. This provides a larger input to the parameter adaptation algorithms to force exploration of different regions than are explored using the steady state 4-2 randomized control tree. Of course, the 5-level tree increases the bit rate or energy required, so in order not to unduly increase the average bit rate or average energy, this additional control tree segment is used for only a short while, 140 samples here, but long enough to cause the adaptation of the parameters to be reinitialized. After the initialization, the 4-2 randomized control sequence from before is used.

Another important part of the Control Tree sequence is the control sequence gain adaptation for each interleaved tree section. These gain adaptation rules are developed in the following section.

*8.2. Control Tree Gain Adaptation*

The control tree output levels and the gain adaptation are major factors in obtaining good overall performance. The branch labels on the control tree can be selected as either uniform or nonuniform, perhaps according to probability density function matched values [32]. The gain adaptation consists of expanding the scaling of the tree values if the last output fell on an outer portion of the amplitude range, contract the gain (scaling) if the last output fell on an inner portion of the amplitude range, and hold the scale fixed if the last output fell on a level between the maximum and minimum dynamic range of the past control tree.

From the four level tree in the multitree in Figure 4, we expand the present scaling by a multiplier of 1.6 if the present output values are on the max positive or max negative outer levels and contract by 0.8 if the present output values are on the inner levels (note that the average performance over all speech utterances examined is not particularly sensitive to these exact expansion/contraction values). There are no intermediate levels on this 4-2 tree where the scaling is held fixed. These multipliers are selected empirically and always can be reoptimized as major changes are made to the rest of the control tree structure. These selected multipliers follow the qualitative rule that expansion should be faster than contraction; however, notice that if all levels of the tree were equally likely, this would lead to an unstable response; however, since for speech, inner levels are much more likely to occur than the outer levels, a larger multiplier for the outer levels makes sense in order to track expansions due to pitch excitations and thus keep the parameter adaptation active.

The above adaptation is for the four-level part of the multitree, and in the 4-2 multitree structure studied here that interleaves the four-level tree component with a binary tree component, we need to determine what adaptation rule to use when the two-level tree occurs. Sometimes the adaptation is accomplished using one symbol, or greater, memory of the past polarities along with the current output sign. We have investigated several multiple stage polarity adaptation rules in addition to simply not updating the gain/scaling when the 1 bit tree is used.

Whenever, the two level tree occurs, we do not have the distinction of an outer and inner level, so we only have polarities to observe and adjust the control sequence gain. We considered the most recent three, four, and five sample polarities to guide how to increase or decrease the control gain when the two-level tree occurs. We studied the performance of the reinforcement learning method for all possible polarity sequences coupled with some experimental gain assignments based on the four utterances F1, F2, M1, and M2. As the memory increased from three to four to five, some slight increases in PESQ-MOS were produced, so we chose the five polarity memory gain policy going forward.

A five sample polarity memory adaptation according to the polarities of the past five control tree output symbols, including both four and two-level tree polarities, is shown in Figure 7. This strategy produced the control sequence gain for the 40-sample 4-2 randomized control sequence in Figure 5 that achieved the best PESQ-MOS performance for the utterances F1, F2, M1, and M2. A gain of 1.0 keeps the tree scaling the same, while a gain greater than 1.0 expands the scale and a gain less than 1.0 contracts the scale; therefore, for what is labeled Case 1, if five polarities of the control value are the same, either all positive or all negative, the control gain when the two level tree occurs is scaled up by 1.26; however, for Case 7 with the last five polarities as shown in the table, the control gain for a two level tree is scaled by 0.80. Note that since the order of the 4-2 tree segments are randomized as shown in Figure 5, the appearance of two or more four-level trees in a five memory segment can greatly impact whether we need to expand or contract or keep the same scaling for a two level segment.

| Case Index (n) | 1 | 2 | 3 | 4 | 5 | 6 | 7 | 8 | 9 | 10 | 11 | 12 | 13 | 14 | 15 | 16 |
|---|---|---|---|---|---|---|---|---|---|---|---|---|---|---|---|---|
| Excitation Polarity Sequence | + − | − + | + − | + − | + − | − + | + − | + − | + − | − + | + − | + − | + − | − + | + − | + − |
| | + − | + − | + − | − + | + − | + − | + − | − + | + − | + − | + − | − + | + − | + − | + − | − + |
| | + − | + − | − + | + − | + − | + − | − + | + − | + − | + − | − + | + − | + − | + − | − + | + − |
| | + − | + − | − + | + − | − + | + − | + − | − + | + − | − + | + − | + − | − + | + − | + − | − + |
| | + − | + − | − + | + − | − + | + − | + − | − + | + − | − + | + − | + − | − + | + − | + − | + − |
| Gain | 1.26 | 1.34 | 1.00 | 1.00 | 1.22 | 0.82 | 0.80 | 1.30 | 1.00 | 0.84 | 1.00 | 1.00 | 1.00 | 0.80 | 0.76 | 1.00 |

**Figure 7.** Control tree gain adaptation for 2-level tree segments.

In addition to having a larger outer level, the 5 level control tree segment also has a more aggressive scaling method. For the 5-level control tree gain adaptation, the control gain is decreased if the zero level is chosen, but the control gain increases if the other two levels occur. The increase for the outer control level is much larger than for the inner level. This approach helps the control energize the parameter adaptation algorithms and excite the System Model to explore additional system states.

## 9. *L* Step Lookahead Cost Function

The approximate Cost-to-Go, $\sum J_k = \sum [s(k) - x(k)]_w^2$, is the sum of perceptually shaped squared errors in the reconstruction, which can be evaluated only at some finite search depth in order not to incur too high a delay that would prevent real time speech coding. In general, however, it is desirable to search as far into the future as possible for the best control sequence although the length of the utterances to be coded can be 40,000 samples easily. For the 4-2 control tree at a search depth of only $L = 10$, a full search of the tree to depth $L = 10$ would explore $4^5$ times $2^5$ or 32,768 paths through the control tree. Even this is clearly impossible, particularly considering that all of the parameter adaptation algorithms would need to be performed over each path.

As a consequence, we adopt the *M* algorithm from channel coding that extends only the *M* best paths to depth *L* at each time instant [33]. Out of these paths, we find the best path and release only the first control value in the best path, and then the *M* best paths starting from this first step are extended to depth *L* again and the process is repeated. This single control symbol release rule is not optimal but it simplifies the implementation. Further, if the number of control symbols released is too close to the full search depth,

$L = 10$ here, then the samples near the end have not had the benefit of sufficient lookahead. A variable symbol release rule may perform better but these have not been explored as yet. A readable development of the $M$ algorithm is available in Anderson and Mohan ([33], pp. 300–306).

However, constraining the number of paths (to $M$) extended after each control value achieving the minimum depth $L = 10$ search of the control tree may cause important paths not to be explored. So, it is important to try and keep the dual control pursuing both exploration and exploitation in other ways.

## 10. Exploitation and Exploration

Using the control sequences defined in Section 8 and their corresponding control gain policies, we can study the effects of exploitation and exploration in the choice of the control policy. Before we focus on the control sequences, however, we need to specify all of the other system details. The System Model to be driven by the dual control sequences is given by Equation (4). The AR coefficient adaptation is accomplished by the least squares lattice algorithm shown in Figure 2 and delineated in Section 5.1, wherein the MA coefficient adaptation is specified by Equation (16), and the quasi-periodic excitation is adapted as in Section 5.2.

The perceptual weighting is performed as described in Section 6 and the approximate Cost-to-Go, $\sum J_k = \sum[s(k) - x(k)]_w^2$, is calculated as in the section on the $L$ Step lookahead, Section 9, using the $M$ control tree search combined with the single control symbol release rule. All of the speech samples used in this paper are taken from the ITU-T P.501 standard [31].

We see from Figure 1 and the details of the adaptation algorithms that the control sequence performs the dual roles of exploration, driving the adaptation of the system parameters, and exploitation, cautiously driving the System Model to track the input speech measured from the Environment.

With the System Model and adaptation algorithms specified, we now consider the performance of the reinforcement learning method using the excitation of the System Model consisting of the 40 sample randomly interleaved four-level and two-level multitree segments as shown in Figure 5 combined with the expansion of the scaling by 1.6 for outer levels and contraction by 0.8 for inner levels for the four-level tree and no scale adjustment for the two-level tree. The performance is measured by the Value Function described in Section 7.

The results are shown in the first row of Table 1 for the speech utterances Female 1 (F1), Female 2 (F2), Male 1 (M1), and Male 2 (M2). In this row, the average PESQ-MOS is 3.438, which although it is less than the desired target of 3.5, is the best performance ever achieved by a tree structured speech codec at the low average bit rates shown.

**Table 1.** PESQ-MOS value function results for three control policies/full utterances.

| Control | Value Function | F1 | F2 | M1 | M2 | Avg | Std Dev |
|---------|---------------|------|------|------|------|------|---------|
| Random 4-2 | PESQ-MOS | 3.377 | 3.376 | 3.443 | 3.556 | 3.438 | 0.073 |
|  | Rate (kbits/s) | 5.92 | 5.57 | 4.09 | 5.41 | 5.25 |  |
| Random 4-2 plus 5-level | PESQ-MOS | 3.425 | 3.515 | 3.544 | 3.569 | 3.513 | 0.054 |
|  | Rate (kbits/s) | 6.03 | 5.66 | 4.15 | 5.49 | 5.33 |  |
| Random 4-2, 5-level, 5pol | PESQ-MOS | 3.471 | 3.562 | 3.584 | 3.599 | 3.554 | 0.050 |
|  | Rate (kbits/s) | 6.03 | 5.66 | 4.15 | 5.49 | 5.33 |  |

To further explore the reinforcement learning approach in Figure 1, we compare this result to inserting a 140 sample five-level control sequence at the beginning of each Voiced speech segment before employing the 40 sample random sequence with the previous adaptation for the four level and no adaptation for the two-level tree. The reason for inserting this more impulse-like excitation at the beginning of each Voiced segment is

to force the control to explore other sequences and larger excitations that will force the parameter adaptation algorithms out of any conservative behaviors. The Value Function results for this control policy are shown in the second row of Table 1.

We further compare to having a 140 sample five-level control sequence at the beginning of each Voiced speech segment before employing the 40 sample random sequence with the previous adaptation for the four-level tree segment and a five-polarity memory gain adaptation for the two-level tree segments as shown in Figure 7. The reason for including gain adaptations during the two-level sequences is to try to impose some persistent excitation on the control sequences and thus include additional randomness as suggested in [10]. The Value Function results for this control policy are given in the third row of Table 1.

We see from Table 1 that there is an increase in the Value Function as we proceed from the random control sequence with no two-level gain adaptation, through the insertion of a five-level control excitation at the beginning of each Voiced speech segment, to the combination of the five-level sequence and 40 sample random control sequence with gain adaptation when the two-level tree segments occur. The gain in PESQ-MOS is about 1.5 standard deviations above the random 4-2 control sequence alone. Additionally, the average PESQ-MOS is just above 3.5. We view the five-level sequence and gain adaptation during the two-level segments as accomplishing additional exploration in addition to the exploitation role of driving the System Model.

The results in Table 1 are for speech utterances that are about 50 percent silence or unvoiced, which are approximated by what is called VAD/CNG, and during which the reinforcement learning methods are not used. To understand the performance produced by reinforcement learning, we need to evaluate the Value Function only on the Voiced segments where reinforcement learning is implemented.

To set up this comparison, we extract the reconstructed speech produced by our reinforcement learning system for the voiced portions of the utterances and concatenate them for separate evaluation. We then extract the corresponding Voiced portions of the original sequences, concatenate them, and code these sequences using the AMR-NB codec at its maximum rate of 12.2 kbits/s. Since our codec operates at 1.5 bits/sample during Voiced segments, at 8000 samples/s, we have a natural comparison at the same rate or the same control energy. The PESQ-MOS results on our system for the three excitations and associated control gain adaptations are presented in Table 2 along with the PESQ-MOS results for using the AMR-NB codec at 12.2 kbits/s on the concatenated voiced sequences.

**Table 2.** PESQ-MOS value function results for three control policies and AMR/voiced only.

| Control | Value Function | F1 | F2 | M1 | M2 | Avg | Std Dev |
|---|---|---|---|---|---|---|---|
| Random 4-2 | PESQ-MOS | 3.646 | 3.581 | 3.784 | 3.699 | 3.678 | 0.074 |
| | Rate (kbits/s) | 12 | 12 | 12 | 12 | 12 | |
| Random 4-2 plus 5-level | PESQ-MOS | 3.747 | 3.798 | 4.022 | 3.793 | 3.840 | 0.107 |
| | Rate (kbits/s) | 12.24 | 12.24 | 12.24 | 12.24 | 12.24 | |
| Random 4-2, 5-level, 5pol | PESQ-MOS | 3.766 | 3.827 | 4.039 | 3.819 | 3.863 | 0.104 |
| | Rate (kbits/s) | 12.24 | 12.24 | 12.24 | 12.24 | 12.24 | |
| AMR, Narrowband | PESQ-MOS | 4.04 | 4.001 | 4.089 | 4.063 | 4.048 | 0.032 |
| | Rate (kbits/s) | 12.2 | 12.2 | 12.2 | 12.2 | 12.2 | |

From these results, shown in Table 2, we see an increase in the Value Function of just under 0.1 to over 0.2 with an average improvement of over 0.15 when we insert the 140 samples of the five-level sequence at the beginning of each voiced segment. Thus, it seems clear that this initialization is producing the desired exploration of new excitations for the parameter adaptations as well as for the System Model input. The five-polarity memory gain for the two-level control sequences bumps the Value Function up slightly more, thus implying persistent excitation; however, perhaps the most important result evident from Table 2 is that the PESQ-MOS values are now above 3.5 and that PESQ-MOS

values in the 3.7 and above range implies near transparent voice quality and intelligibility with little distortion.

We see from the last two rows of Table 2 that the AMR-NB 12.2 kbits/s coding mode consistently achieves a PESQ-MOS of 4.0, or Very Good, which is better than the performance of our reinforcement learning based approach on average. For Male 1, almost identical performance is achieved.

Since we have used the two Female, F1 and F2, and two Male sentences, M1 and M2, in Tables 1 and 2 in our research optimizations, we now consider additional female and male sentences outside of the optimization set to evaluate the performance of reinforcement learning. To this end, Table 3 shows the performance of our three control policies and the comparison to AMR-NB for female speakers outside of the optimization set and voiced segments only, which are the regions where reinforcement learning is used. We see that there is a slight upward trend in the PESQ-MOS, but just less than one standard deviation gain, when we insert the 140 samples of the five-level sequence at the beginning of each voiced segment and then including the 5 polarity memory control gain; however, the additional increase in PESQ-MOS achieved by using the five-polarity memory for the control gain is certainly negligible in terms of audible perception. Further, the performance change when we insert the 140 samples of the five-level sequence at the beginning of each voiced segment is not uniform across all sentences. In comparison to AMR, AMR outperforms our reinforcement learning approach on average, but for F9 and F11 our codec performs better.

**Table 3.** PESQ-MOS value function results for three control policies and AMR/voiced only–additional female sentences.

| Control | F4 | F5 | F6 | F7 | F8 | F9 | F10 | F11 | Avg | StD |
|---|---|---|---|---|---|---|---|---|---|---|
| Random 4-2 | 3.826 | 3.543 | 3.597 | 3.577 | 3.602 | 3.54 | 3.634 | 3.669 | 3.624 | 0.087 |
| Random 4-2 plus 5-level | 3.878 | 3.769 | 3.632 | 3.512 | 3.721 | 3.677 | 3.624 | 3.787 | 3.700 | 0.106 |
| Random 4-2, 5-level, 5pol | 3.861 | 3.772 | 3.639 | 3.514 | 3.733 | 3.68 | 3.638 | 3.79 | 3.703 | 0.102 |
| AMR, Narrowband | 3.978 | 3.96 | 3.721 | 3.818 | 3.923 | 3.634 | 3.954 | 3.697 | 3.836 | 0.128 |

Similar performance results are shown in Table 4 for male speakers. The behavior of these results is similar to those for the female speakers in that for M10 we see that the addition of the five-level sequence at the beginning of each voiced segment and then including the five-polarity memory control gain actually reduces the PESQ-MOS by 0.02. Further, in comparison to AMR, we see that AMR again performs better on average, but for M9 individually, reinforcement learning performs better.

**Table 4.** PESQ-MOS value function results for three control policies and AMR/voiced only–additional male sentences.

| Control | M4 | M5 | M6 | M7 | M8 | M9 | M10 | Avg | StD |
|---|---|---|---|---|---|---|---|---|---|
| Random 4-2 | 3.645 | 3.684 | 3.856 | 3.662 | 3.796 | 3.848 | 3.828 | 3.760 | 0.086 |
| Random 4-2 plus 5-level | 3.806 | 3.792 | 3.877 | 3.723 | 3.801 | 3.878 | 3.793 | 3.810 | 0.050 |
| Random 4-2, 5-level, 5pol | 3.804 | 3.797 | 3.88 | 3.736 | 3.806 | 3.875 | 3.808 | 3.815 | 0.046 |
| AMR, Narrowband | 3.984 | 4.091 | 3.973 | 3.902 | 3.824 | 3.745 | 4.052 | 3.939 | 0.114 |

Another observation for all of the results in all four tables is that the reinforcement learning codecs and AMR show poorer performance for female speakers. Historically, this

is true for analysis-by-synthesis speech codecs that operate in the bit rate range of 8 to 16 kbit/s (roughly). This difference is usually attributed to the fact that female pitch periods are shorter and thus, for fixed-length-block-based coders such as AMR, there are more pitch periods in a frame that are averaged together. For our proposed codec structure, this long term redundancy due to pitch is difficult to track by our current backward adaptive algorithm, so additional pitch pulses in a shorter period of time create tracking difficulties. Note that a difference in performance between female and male speakers may not hold for all speech codec structures. For example, low bit rate codecs that operate at 4 kbits/s and below [34], have very different forms than the analysis-by-synthesis CELP codecs [35]. Further, newer neural-based codecs do not show differences between female and male speakers, but the structure of these speech codecs and their design procedures are quite different than CELP and other low bit rate codecs and their performance cannot be determined from PESQ-MOS [36].

An additional note is that the results for the sentences in Tables 3 and 4 are not as good as in Table 2, as should be expected since the four sentences in Table 2 were used in our codec optimizations. This implies that more extensive optimizations should be performed over larger data sets. This work is in progress, Table 5 contains results for female and male sentences combined.

**Table 5.** Average PESQ-MOS value function for three control policies and AMR/voiced only—male and female sentences outside of the optimization set.

| Control | Average | Standard Deviation |
|:---:|:---:|:---:|
| Random 4-2 | 3.687 | 0.110 |
| Random 4-2 plus 5-level | 3.751 | 0.101 |
| Random 4-2, 5-level, 5pol | 3.756 | 0.098 |
| AMR, Narrowband | 3.884 | 0.132 |

Since the AMR-NB codec is the result of years of standardization meetings with hundreds of engineers and researchers, the current performance advantage is to be expected; however, indications are that the reinforcement learning approach is solid, but optimization of reinforcement learning parameters and control policies should include a wider class of speakers and sentences. The reinforcement learning method introduced here opens up many new research directions as outlined in the Conclusions.

## 11. Conclusions

The approach presented here for speech coding responds to the Environment by exciting the System Model and the associated parameter adaptation algorithms to learn about the state of the Environment to minimize a cost-to-go and produce improved performance as exhibited by the chosen Value Function, PESQ-MOS. The results show that this method of speech coding does indeed exhibit the characteristic of learning by interacting with the environment, as used by Sutton and Barto to define reinforcement learning [1]. To encourage Exploration, an additional excitation is applied at the beginning of every Voiced speech segment to force the parameter adaptation algorithms to explore new trajectories. This control policy does achieve an improvement in the Value Function. Additional randomness is used in the excitation, as suggested by Bertsekas and Tsitisklis [10], to provide a control policy that sustains exploration and keeps the parameter adaptation algorithms from becoming stagnant.

This reinforcement learning approach to speech coding not only achieves the best performance ever by a tree structured speech codec at the low average bit rates shown, but it affords new insights into how to design speech codecs and into the roles of the

several components, namely, the System Model, the parameter adaptation or learning algorithms, the Cost-to-Go, the Control Policy, and the lookahead function. There is much more work to perform to achieve the full promise of this speech coding approach, including different implementations of the lookahead function, better parameter adaptation methods (particularly with respect to the MA part and the long term prediction), the form and the settings in the perceptual weighting, and other Control policies.

**Author Contributions:** Conceptualization, J.G.; software, H.O.; data curation, H.O.; writing—original draft preparation, J.G.; funding acquisition, J.G. All authors have read and agreed to the published version of the manuscript.

**Funding:** This research was funded, in part, by a contract with Raytheon Applied Signal Technology in San Jose, California.

**Data Availability Statement:** The speech utterances used in this work were taken from the ITU-T P.501 set of test utterances found at https://www.itu.int/rec/T-REC-P.501(accessed on 15 June 2022). The narrowband British English utterances for two female and two male speakers were used as F1, F2, M1 and M2.

**Conflicts of Interest:** The authors declare no conflict of interest.

## Abbreviations

The following abbreviations are used in this manuscript:

| | |
|---|---|
| AMR-WB | Adaptive Multirate-Wideband |
| AR | Autoregressive |
| MA | Moving Average |
| PESQ-MOS | Perceptual Evaluation of Speech Quality-Mean Opinion Score |
| RLS | Recursive Least Squares |
| VAD/CNG | Voice Activity Detection/Comfort Noise Generation |

## Appendix A. DPCM and Tree Coding

The reinforcement learning approach introduced in this paper for speech coding builds upon two classic speech coding principles, predictive coding, and delayed encoding. Although much more extensive treatments of these concepts for speech coding are available elsewhere, we provide concise developments here to put the reinforcement learning approach into context. First, we introduce differential pulse code modulation (DPCM) and then add the extension to tree coding.

### Appendix A.1. DPCM

The most common example of predictive coding for speech is classical differential pulse code modulation as shown in Figure A1.

Knowing that speech samples are highly correlated during Voiced speech segments, it is intuitive to use recent past speech samples to predict the next sample, thus reducing the dynamic range of the signal to be quantized and coded; however, the input speech samples are not available at the Decoder so past reconstructed samples must be used to form the predictor, as shown in Figure A1 [32,37]. The predicted value is subtracted from the input sample and the prediction error is passed through the scalar quantizer. Binary codes are assigned to the several quantizer levels and these codewords are sent to the Decoder. A characteristic of DPCM is that the Encoder includes a "copy" of the Decoder, so the Encoder and Decoder operate in concert unless there are bit errors during transmission over a channel. The predictor and quantizer are usually adaptive in some fashion, and there are extensive treatments in the literature of many possible designs [32,37,38], including using least squares lattice algorithms for the short term predictor [27].

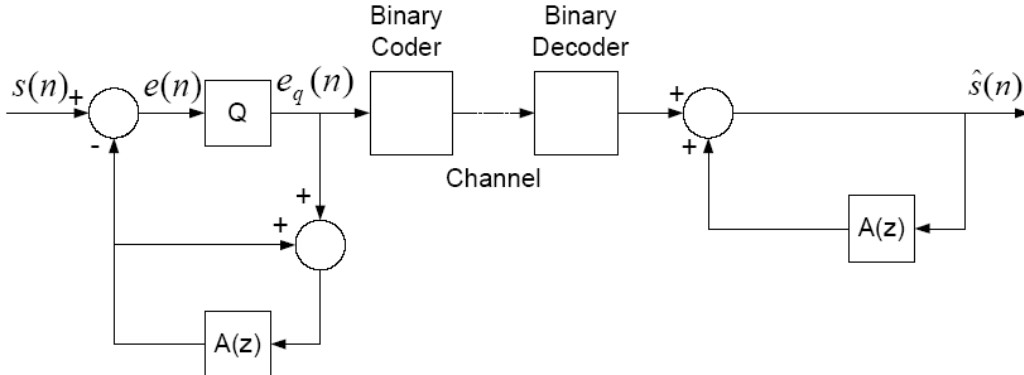

**Figure A1.** Classical Differential Pulse Code Modulation (DPCM).

*Appendix A.2. Tree Coding*

Based on the DPCM architecture, it is possible to formulate a delayed encoding approach wherein a DPCM system is used to generate current and future predictions, quantizer values, and reconstructed outputs for some lookahead number of samples, and thus, since the quantizer has a set of discrete levels, a tree structure is generated to the chosen depth. A distortion can be calculated for each path through the tree and the best path chosen, then that path is sent to the Decoder for reconstruction.

However, the greatest impetus for tree coding came from a rate distortion theory motivation [2,3], wherein the quantizer based tree could be replace with randomly populated trees and the number of symbols per branch, limited to one sample per branch by a scalar quantizer, could be greater than one, thus producing fractional rate trees and codes [39]. Early work in this area use time domain unweighted squared error as the fidelity criterion and nonadaptive predictors, so competitive speech coding was not achieved; however, later work incorporating adaptive algorithms, fractional rate trees, and perceptually weighted distortion measures did produce promising results [4]. While recent work on tree coding, motivated by greatly reducing complexity and achieving very low latency using simple predictive algorithms have produced good performance at higher average bit rates [6], research at much lower average rates have not achieved competitive performance compared to the block-based code excited CELP codec designs. The reinforcement learning method presented in this paper points a way forward.

**Appendix B. Pitch Lag Adaptation**

The approach used here is a combination of the methods by Ramachandran and Kabal [28] and by Pettigrew and Cuperman [26,29]. After the pitch lag is initialized by calculating a block autocorrelation function from previous outputs of the pitch excitation, then

$$\begin{pmatrix} \beta_{-1} \\ \beta_0 \\ \beta_{+1} \end{pmatrix} = \begin{pmatrix} (1+\mu)R_{ee}(0) & R_{ee}(1) & R_{ee}(2) \\ R_{ee}(1) & (1+\mu)R_{ee}(0) & R_{ee}(1) \\ R_{ee}(2) & R_{ee}(1) & (1+\mu)R_{ee}(0) \end{pmatrix}^{-1} \begin{pmatrix} R_{ee}(d-1) \\ R_{ee}(d) \\ R_{ee}(d+1) \end{pmatrix}, \tag{A1}$$

where $\mu = 0.001$ is the softening factor, which improves filter performance. The stability of the initialized weighting coefficients is determined by the stability test, described in Appendix C.1. If the initialized weighting coefficients are unstable, the stabilization procedure, explained in Appendix C.1, needs to be applied to the initialized weighting coefficients. The coefficients are scaled by a factor $c$. When the weighting coefficients of each block are initialized, the coefficients are recursively adapted sample-by-sample by using the equation:

$$\beta_i(k) = \theta\beta_i(k-1) + \frac{\zeta}{\sigma_u(k)\sigma_e(k)}u(k)e(k-d_k-i), \quad i = -1, 0, +1, \tag{A2}$$

where $\theta$ and $\zeta$, are parameters to be selected, $u(k)$ is the tree control sequence, $\sigma_u^2$ is the estimate of the variance of $u(k)$, and $\sigma_e^2$ is the estimate of the variance of the weighted $e(k)$. The estimated variances are calculated using Equation (18). If the updated weighting coefficients are not stable, then no update is performed on the coefficients. When the pitch lag is either incremented or decremented by one, the weighting coefficients are shifted by one in the appropriate direction. The new coefficient, either $\beta_{-1}$ or $\beta_{+1}$ is computed as a constant fraction of $\beta_0$, typically 0.67. A stability test is used for the 3-tap weighted periodic excitation to determine whether the predictor is stable or not. If the weighting coefficients are not stable, a stabilization procedure is required. If the updated coefficients are unstable, the pitch lag and the weighting coefficients are re-initialized by Equations (A1) and (A2), respectively.

**Appendix C. Pitch Stability Test**

The three tap long term predictor can be unstable based on the continuous adaptation of the pitch and predictor coefficients. As a result, we employ a well established stability check and a stabilization procedure as described here.

*Appendix C.1. Stability Test*

The stability test is summarized below [26,28,29].
*Stability Test:* Let $a = \beta_{-1} + \beta_{+1}$ and $b = \beta_{-1} - \beta_{+1}$.
1. If $|a| \geq |b|$, the following is sufficient for stability:
   (a) $|\beta_{-1}| + |\beta_0| + |\beta_{+1}| < 1$
2. If $|a| < |b|$, the satisfaction of the two following conditions is sufficient for stability:
   (a) $|\beta_0| + |a| < 1$
   (b) (i) $b^2 \leq |a|$ or
       (ii) $b^2 \beta_0^2 - (1 - b^2)(b^2 - a^2) < 0$

*Appendix C.2. Stabilization Procedure*

There are several ways to stabilize pitch coefficients. One way is to use a scale factor $c$ to re-scale the pitch coefficients $\beta$. After scaling by the factor $c$, the vector of pitch coefficients is $\beta' = c\beta$. *Procedure for determining scale factor $c$:*
1. If $|a| \geq |b|$,

$$c = \frac{1}{|\beta_{-1}| + |\beta_0| + |\beta_{+1}|}. \tag{A3}$$

2. If $b^2 \leq |a|$,

$$c = \frac{1}{|a| + |\beta_0|}. \tag{A4}$$

3. If $b^2 > |a|$,

$$c = \sqrt{\frac{b^2 - a^2}{b^4 + b^2 \beta_2^2 - b^2 a^2}}. \tag{A5}$$

**Appendix D. Voice Activity Detection/Comfort Noise Generation (VAD/CNG)**

We only classify the input speech as Voiced, Unvoiced, or Silence. Both the Unvoiced segments and Silence segments are coded using CNG in the same way. The segments declared to be Voiced speech are passed through the reinforcement learning processing steps as shown in Figure 1.

To make the decision about voice activity, the VAD uses the future Hamming windowed 128 samples. If the block is declared Silence, those same 128 samples are used to calculate the eight linear predictor coefficients used in CNG. These coefficients are different than the ones used in the Perceptual Weighting Filter (PWF). The VAD flag and the Silence Insertion Descriptor (SID) information is then sent to the Decoder. When no voice activity

is detected, the gain or step size adaptation is halted and the gain at the beginning of the VAD/CNG frame is held fixed until voice activity is detected again. A VAD flag and the Silence No Update (SNU) frame is sent to the Decoder.

Voicing classification is performed using the total power of the 128 samples, the power in the high frequency band above 1.75 kHz, and a low power threshold. If the total power is less than the power threshold, Silence is detected. Otherwise the ratio of the power in the high frequency band to the total power is compared to a threshold. If this high frequency band power to total power ratio is greater than a threshold, the speech is classified as Unvoiced. Unvoiced speech is coded using the same CNG method used for Silence. If the speech is not classified either as Silence or Unvoiced speech, it is classified as Voiced and coded with the reinforcement learning algorithms. The VAD/CNG provides a substantial reduction in bit rate but incurs a noticeable drop in the PESQ-MOS value and the perceptual quality. This VAD/CNG method is outside of the reinforcement learning processing and will need to be improved to produce the highest quality speech attainable when combined with the reinforcement learning architecture.

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
