# Peer review of "A Reinforcement Learning Approach to Speech Coding"

_information, doi:10.3390/info13070331_

Round 1

Reviewer 1 Report

The paper investigates the use of reinforcement learning for speech coding and based on sample-by-sample coding approach. Linking reinforcement learning to this type of speech coding is an interesting idea and results on the evaluated speech files are comparable to the well-known Adaptive Multi Rate-Narrowband (AMR-NB) speech coding standard.

While the approach is interesting it would be good to further clarify what the benefits might be. For example, it is mentioned in the abstract that the approach results in lower latency to the most popular speech codec (assumed to the AMR-NB codec evaluated in the paper) however it would be good to more clearly compare and contrast the latency of the two approaches.

As a general comment, some parts of the paper are hard to follow and there are various sections that appear to lack references or brief explanations to clarify what parts of the approaches are new and what parts are based on existing approaches. In contrast, a recent paper [a] co-authored by the first author of this paper has some clearer descriptions of some of relevant background such as Differential Pulse code Modulation (DPCM) speech coding, the ITU-T G.727 standard and references where some of the key equations have been described e.g. the short-term predictor, adaptive pitch predictor etc. Please consider revising such sections to make them clearer.

In Section 2, it might be good to include a brief review of DPCM speech coding and recursive tree coding of speech.

Section 4 describes a well-known system model used in DPCM speech coding but it would help to include the system block diagram (e.g. similar to one used in [a]) and clarify any specific aspects of the model that are different. It would appear that one difference is the interpretation of u(k) as the control sequence within the reinforcement learning framework but please clarify.

In Section 5, please add some references for some of the sub-sections that appear to describe well-known and standardised methods such as the Recursive Least Squares (RLS) Lattice algorithm, (16) and (17). In Figure 2, it wasn’t clear if some of the symbols were described in the text or alternative symbols were used e.g. rbk(n), xsp(n) etc.

In Section 6, it is mentioned that the model coefficients used in (20) are calculated using a lookahead of 200 samples. Please clarify how this affects the latency of the system.

Section 7 mentions that the reinforcement method is only used for voiced speech. Please clarify what is meant by “voiced speech” here. Speech can be typically classified as voiced, unvoiced or silence but it is unclear here if “voiced speech” means “non-silent” periods of active speech as detected by Voice Activity Detection (VAD) or periods that are neither unvoiced or silence. Later on page 13 it seems to say that comfort noise is used for both silence and unvoiced parts of the utterances however Appendix C seems to be describing standard silence detection.

Section 8 is a bit hard to follow. It may help if there was a short introduction to tree coding of speech. It is mentioned that PESQ-MOS evaluated over all sentences is used to find the best 40 sample control sequence. But it is not so clear how this is achieved as it would seem dependent on the chosen speech files and implies a training phase using speech that is not used in testing. Please clarify.

In Section 8, please further explain Figure 7 as it is not so clear what these polarity sequences mean.

In Section 9, is the M, L algorithm related to the multimode tree coding of speech approach? Please clarify.

 Is it possible to make some comments or present results comparing the performance with the multimode tree coding approach of [a]? It would good to contrast with this other approach by the first author.

Appendices A, B and C appear to be describing the details of some standardised approaches for pitch lag adaptation, pitch stability testing and VAD/CNG. Please clarify this and provide references as appropriate.

[a] Li, Y.-Y.; Ramadas, P.; Gibson, J. Multimode Tree-Coding of Speech with Pre-/Post-Weighting. Appl. Sci. 202212, 2026. https://doi.org/10.3390/app12042026

Reviewer 2 Report

The authors present preliminary work on a new paradigm for speech coding by using the reinforcement learning framework to code voiced segments of speech and recasting some of the classical ideas into this, also classical framework, that is nowadays producing interesting results. Overall, the work is sound but the experimental part needs a more thorough evaluation. 

In general, there are a number of decisions that need a proper justification and validation. For example, in eq (16) ad hoc coefficients are chosen or in eq (19) typical values for \nu and the autocorrelation fraction, the number of AR coefficients, etc.  A better definition for J_k should be given.

Figure 3 is regarded as 'the input speech spectrum' (p7, l224) when it is only the envelope of the spectrum.

Regarding the experimentation, the fact that only 2 female and 2 male voices are considered, makes the results weak and poor. Since only PESQ is used (and no subjective evaluation), it should be feasible to use many more speakers and sentences. Also standard deviation values should be provided to statistically profile the results. Also the gender influence should be evaluated and a hypothesis about the consistently worse results for the female samples should be provided. Since there are only four samples, it is imposible to know if this is a systematic bias or just due to the specific choice of samples.  

Replicability is not possible and it would be interesting to openly release the implementation. 

The paper is very well-written. However, there are a couple of typos: 

p15, l518: 'Le' at the end of the sentence should be deleted.

p15, l523: The acronym 'PWE' should be explained.

9 out of the 24 references are self-references although I do not find them inappropriate. 

Round 2

Reviewer 1 Report

The new version of the paper has addressed all of my previous comments very well. Various sections have been clarified. The further explanation of Figure 7 with reference to examples in the table and the addition of sections in the Appendices to provide further background to ADPCM coding are welcomed. This has made the paper easier to follow and I have no further comments.

Author Response

Thank you for your comments.  We are pleased to have responded to all of you comments and suggestions.

Reviewer 2 Report

The authors have answered satisfactorily to all my suggestions except the one regarding further experiments with more speakers which, they have provided, but whose results raise important questions.

First, it is absolutely necessary to provide standard deviation values to be able to appreciate how significative the findings are, statistically speaking.

Second, the differences in the nomenclature in tables 1 and 2, on the one hand and 3 and 4, on the other, must be explained (why now using Case 1, 2..? Aren't they the same control policies than before?). Moreover, it would be better to include all the samples in Tables 3 and 4 into the analysis of table 2 (at least the mean and standard deviations).

Third and more important, the conclusions and interpretation of the experiments should be rewritten for consistency:

1. in lines 494-496, a relative difference of 4.4%  in the Value Function is considered a “substantial jump”. This difference does not translate into Tables 3 and 4 that show a difference of 2.1% and 1.3%

2. but in lines 503-504, a relative difference of 4.8% is considered "slightly better".

3. in lines 523-526, the difference between male and female performance is considered "slightly poorer" for females but in Tables 3 and 4, this difference is very similar than the one that had been earlier considered "substancial" for table 2.

Fourth, the explanation provided for the differences in performance for male and female speakers in lines 524-526 is not satisfactory. Please, provide references on the influence of the worse pitch estimation in females on the quality of the codecs. A discussion should be provided since not all the codec technologies exhibit this problem. See for example, the comparison with neural-based codecs in:

W. A. Jassim, J. Skoglund, M. Chinen and A. Hines, "Speech Quality Factors for Traditional and Neural-Based Low Bit Rate Vocoders," 2020 Twelfth International Conference on Quality of Multimedia Experience (QoMEX), 2020, pp. 1-6, doi: 10.1109/QoMEX48832.2020.9123109.

Round 3

Reviewer 2 Report

The authors have addressed all my concerns and I now think that the text accurately describes the importance and extent of the results.

Author Response

Thank you for your comments.  We are pleased to have addressed all of your comments and concerns.  The manuscript is much improved.